# Assessment of Burn Severity and Monitoring of the Wildfire Recovery Process in Mongolia

Battsengel Vandansambuu [1,2], Byambakhuu Gantumur [1,2,*], Falin Wu [3], Oyunsanaa Byambasuren [4], Sainbuyan Bayarsaikhan [1,2], Narantsetseg Chantsal [1,2], Nyamdavaa Batsaikhan [1], Yuhai Bao [5], Batbayar Vandansambuu [1,2] and Munkh-Erdene Jimseekhuu [1,2]

1 Department of Geography, School of Arts and Sciences, National University of Mongolia, Ulaanbaatar 14200, Mongolia; battsengel@num.edu.mn (B.V.); sainbuyanb@num.edu.mn (S.B.); narantsetsegch@num.edu.mn (N.C.); nyamdavaab@num.edu.mn (N.B.); batbayar@outlook.com (B.V.)
2 Laboratory of Geo-Informatics (GEO-iLAB), Graduate School, National University of Mongolia, Ulaanbaatar 14200, Mongolia
3 SNARS Laboratory, School of Instrumentation and Optoelectronic Engineering, Beihang University, Beijing 100191, China; falin.wu@buaa.edu.cn
4 Regional Central Asia Fire Management Resource Center, National University of Mongolia, Ulaanbaatar 14200, Mongolia; oyunsanaa@num.edu.mn
5 Inner Mongolia Key Laboratory of Remote Sensing and Geographic Information Systems, Inner Mongolia Normal University, Hohhot 010022, China; baoyuhai@imnu.edu.cn
* Correspondence: byambakhuu@num.edu.mn; Tel.: +976-99994813

**Abstract:** Due to the intensification of climate change around the world, the incidence of natural disasters is increasing year by year, and monitoring, forecasting, and detecting evolution using satellite imaging technology are important methods for remote sensing. This study aimed to monitor the occurrence of fire disasters using Sentinel-2 satellite imaging technology to determine the burned-severity area via classification and to study the recovery process to observe extraordinary natural phenomena. The study area that was sampled was in the southeastern part of Mongolia, where most wildfires occur each year, near the Shiliin Bogd Mountain in the natural steppe zone and in the Bayan-Uul sub-province in the forest-steppe natural zone. The normalized burn ratio (NBR) method was used to map the area of the fire site and determine the classification of the burned area. The Normalized Difference Vegetation Index (NDVI) was used to determine the recovery process in a timely series in the summer from April to October. The results of the burn severity were demonstrated in the distribution maps from the satellite images, where it can be seen that the total burned area of the steppe natural zone was 1164.27 km$^2$, of which 757.34 km$^2$ (65.00 percent) was classified as low, 404.57 km$^2$ (34.70 percent) was moderate-low, and the remaining 2.36 km$^2$ (0.30 percent) was moderate-high, and the total burned area of the forest-steppe natural zone was 588.35 km$^2$, of which 158.75 km$^2$ (26.98 percent) was classified as low, 297.75 km$^2$ (50.61 percent) was moderate-low, 131.25 km$^2$ (22.31 percent) was moderate-high, and the remaining 0.60 km$^2$ (0.10 percent) was high. Finally, we believe that this research is most helpful for emergency workers, researchers, and environmental specialists.

**Keywords:** wildfire; burn severity; vegetation recovery; Sentinel-2; Eastern Mongolia





## 1. Introduction

Wildfires are natural phenomena that have been occurring for centuries; however, in recent years their severity and frequency have increased significantly, posing substantial challenges to ecosystems and human communities [1–4]. Wildfires have emerged as a global concern due to their destructive potential and wide-ranging consequences. Wildfires have been increasing over the last few years due to climate change, forming different natural zones based on location.

Remote sensing using satellites is the most important method for mapping natural disasters, including wildfires, floods, storms, and other extreme weather phenomena, as it acquires pre-disaster and post-disaster data [5]. Remote sensing satellite data are used for assessing damage and environmental conditions post-disaster and for analyzing risk estimation and vulnerability pre-disaster.

Understanding and assessing fire severity can be considered a form of wildfire recognition research. It is especially important to reduce, prevent, prepare, and respond to the damage wildfires cause. The increase in wildfires in recent years is due to climate change. Wildfires mostly cause damage to human environments, leading to infrastructure and economic losses, and cause serious damage to vegetation and wild animal environments while leading to changes in the evolution of nature and ecology patterns [6].

Many studies from the wildfire severity study field were reviewed. The studies were carried out using different satellite images, including optical [7–10], thermal [11,12], lidar [13], and synthetic aperture radar (SAR) [11,14,15] satellite images. Most of them used optical satellite images obtained from MODIS data, Sentinel-2, Landsat series images, and KOMPSAT-3A [16], which uses Shortwave Infrared (SWIR) bands for the calculation. The SAR images were from Sentinel-1, ALOS-2 [17], and PALSAR-2 [18]. There was another interesting paper that evaluated the sensitivity of full-waveform LiDAR data to estimate the severity of wildfires using a 3D radiative transfer model approach [19]. However, all of these studies using LiDAR, SAR, and thermal satellite images made estimations with the optical satellite images by comparing the burned severity area and recovery processes.

This study of post-fire recovery concepts is very important to understand this phenomenon. There are some different approaches to define recovery processes, including utilizing the strong performance and suitability of the post-fire stability index [20]; random forest classification models that use the fire severity classes (from the Relativized Burn Ratio (RBR)) as a dependent variable and 23 multitemporal vegetation indices [21]; post-fire stream water responses observed in watersheds [22]; multiple factors of a forest's recovery rate post-wildfire such as fire severity, tree species characteristics, topography, hydrology, soil properties, and climate [23]; and a composing study based on a Tasseled Cap linear regression trend in a post-wildfire study site [24].

Despite extensive research on wildfire severity, there are still gaps in these studies, especially in Mongolian-specific geographic regions under certain climatic conditions. This study aims to address this gap by focusing on completing a severity assessment of wildfires in the Eastern Mongolian region, which is characterized by unique topographical features and a diverse range of vegetation types. By investigating the relationship between fire behavior, fuel characteristics, and climatic factors, we seek to enhance the accuracy and effectiveness of wildfire severity assessments in this region.

Other issues and different phenomena occur in different areas within different natural zones around the world. We collected and reviewed a few studies from different study areas, including Siberia, Russia [23,25,26], Indonesia [27], Canada [28,29], Australia [18,30–32], Spain [33], Portugal [13], the Mediterranean [7,34–37], Turkey [1,4], Greece [2,3], China [10,38–41], California and Alaska [42–44], the US [45–49], Peru [14], Iran [50], Bolivia [51], the Amazon of Brazil [52] and India [53]. The wildfire studies from each country had their own characteristics. In particular, determination via the mapping of burn severity with the NDVI in the Turkish study [1] was relevant to our study. The difference was found just between pre-wildfires and post-wildfires.

Mongolian wildfires have increased due to climate change, which intensively influences natural disasters and environmental conditions. Many studies have looked into this, such as by determining the fire history from tree rings for the potential of relationships between climate change, fire, and land uses [54,55]; the effects of wildfire on runoff generation processes in mountainous forest areas [56]; wildfire and climate change's effect on permafrost degradation [57]; and wildfire risk mapping for protected areas [58]. There are other wildfire study cases that have been carried out for the Mongolian Plateau, such as a study that identified drivers and spatial distributions to predict wildfire probability [59],

one that explored the growing season [60], one that performed an analysis of climate fire relationships and an evaluation of the spatial change characteristics [61], and one that analyzed the spatiotemporal wildfire pattern using satellite images [62]. Some researchers determined the cost effects of monitoring vegetation changes in steppe ecosystems [63]. Most wildfire impact cases occur across borderland areas between Mongolia, Russia, and China [64,65].

The purpose of this study is to monitor the occurrence of fire disasters as a result of Sentinel-2 satellite imaging technology by carrying out a classification of the burned area and determining the recovery process effects in Eastern Mongolia. This study is based on the recovery effects and processes used in our previous wildfire projects and a few direct Mongolian papers that have covered different study areas, including forest [66] and steppe [6] wildfires.

## 2. Analysis Methods for Wildfire Monitoring

Analysis methods for wildfire monitoring include two steps: determining the spectral response for satellite images and determining the statistical analysis response. The spectral response for satellite images is based on the NBR, differenced NBR (dNBR), RBR, and NDVI indices. The statistical analysis response is based on regression analysis.

### 2.1. Spectral Response for Satellite Images

It is important to include sensor spectral information in satellite images. Remote-sensing-based burn severity indices have been developed and used due to their simple computation and direct applications for image processing. Near-infrared (NIR) and Short-wave Infrared (SWIR) bands are useful for this study.

Spatial and multitemporal NBR images are estimated as the proportion of the difference and sum of the NIR and SWIR bands (Band 9 and Band 12 of Sentinel-2, respectively), which is demonstrated in Equation (1) (Figure 1). The NIR reflectance decreases due to vegetation loss, and the SWIR reflectance increases due to a reduction in canopy humidity and shade [67].

$$\text{NBR} = \frac{(\text{NIR} - \text{SWIR})}{(\text{NIR} + \text{SWIR})} \tag{1}$$

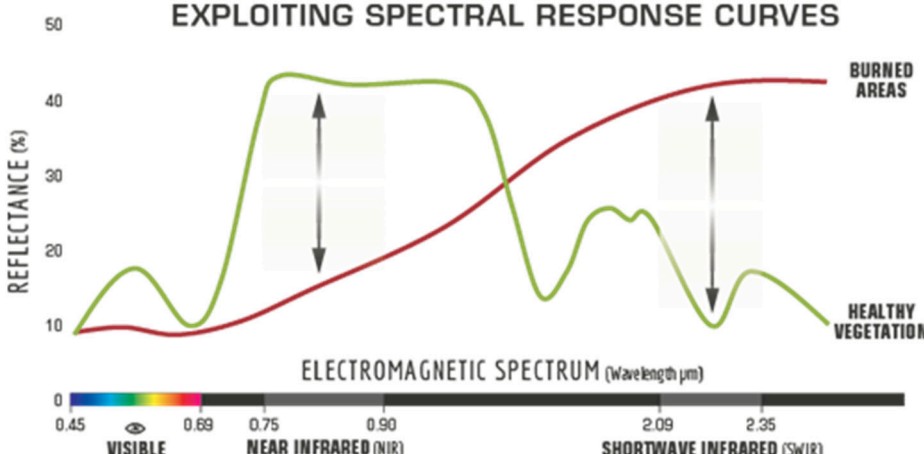

**Figure 1.** Healthy plant species reflect more energy in the NIR band, and burned areas reflect more energy in the SWIR band. This spectral characteristic is useful for detecting burned areas and vegetation recovery over land. Source: US Forest Service.

The dNBR is estimated by pre-fire and post-fire NBR values (Equation (2)). It considers the difference between the burned area and the unburned area.

$$\text{dNBR} = \text{NBR}_{\text{pre−fire}} - \text{NBR}_{\text{post−fire}} \tag{2}$$

The RBR is a variant of the dNBR that considers the relative amount of pre- to post-fire change by dividing the dNBR by the pre-fire NBR value. This index was proposed to remove the bias due to the pre-fire vegetation type and density [68]. Equation (3), which determines the RBR index, comes from a combination of Equations (1) and (2).

$$RBR = \frac{(dNBR)}{(NBR_{pre-fire} + 1.001)} \tag{3}$$

Remotely sensed vegetation indices have also been used to analyze post-fire recovery. The NDVI uses Sentinel-2 imagery to estimate and monitor vegetation recovery and growth after successive fires.

$$NDVI = \frac{(NIR - RED)}{(NIR + RED)} \tag{4}$$

The geographical scales (i.e., geometry, pixel size, and projection) of all estimated variable files should be at the same scale for the next step of determining the statistical analysis response.

### 2.2. Statistical Analysis Response

Regression analysis and scatter plots were used for the statistical analysis of responses. In this case, we tried to determine the phenomenon where the recovery processes are dependent on the NBR from the start of the wildfire to the end of recovery.

Correlation and regression analyses can be conducted with this statistical method to obtain the relationship between the indicator factors [69]. Pearson's correlation coefficient (*r*) indicates the correlation between two variables that are determined by

$$r_{xy} = \frac{\sum\limits_{i=1}^{n} (x_i - \overline{x})(y_i - \overline{y})}{\sqrt{\sum\limits_{i=1}^{n} (x_i - \overline{x})^2} \sqrt{\sum\limits_{i=1}^{n} (y_i - \overline{y})^2}} \tag{5}$$

where $n$ is the sample size; $x_i$ and $y_i$ are the individual sample points indexed with $i$; and $\overline{x} = \sum_{i=1}^{n} x_i$ (the sample mean), which is analogous for $\overline{y}$.

The regression analysis was carried out with linear regression and scatter plot graphs that used intercept and slope. This demonstrates that scattering distributions change shapes and locations and describe the phenomena that we want to see. Correlation and regression analyses were used to determine the relationships between the NBR and NDVI during the recovery process.

## 3. Wildfire Case Study

### 3.1. Study Areas

The first sampled wildfire happened at Shiliin Bogd Mountain on 17–20 April 2021. This mountain is located in Eastern Mongolia (45°20′–45°40′ N and 114°20′–115°20′ E; Figure 2(1)) in the Sukhbaatar province, which is on the border between Mongolia and China. The burned area was calculated to be 6879.67 km$^2$ at an elevation between 1300 and 1800 m. The average annual temperature of the whole range is 1.5–1.7 °C (the mean maximum is 30 °C in July and the mean minimum is −32.5 °C in January), and the average annual precipitation is 200.6 mm. Precipitation follows a bimodal distribution, with maxima in June–September and November–February. The rainiest months are July and August, with 150 mm of rain on average, and the snowiest months are December and January, with 40 mm of snow on average. Additionally, the driest periods occur from March to May and from September to November.

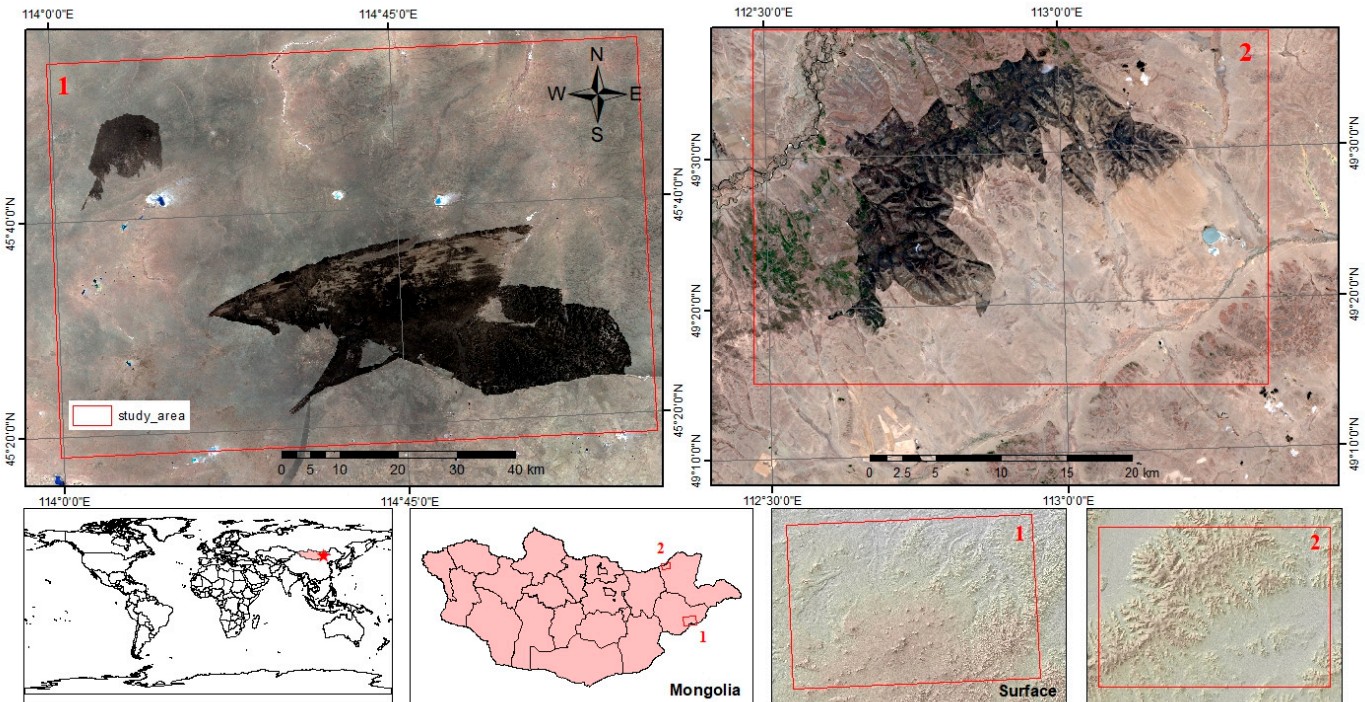

**Figure 2.** The sampled burned areas were at Shiliin Bogd Mountain and the sub-provinces of Bayan-Uul and Bayandun in Eastern Mongolia. Sentinel-2 satellite imagery from 20 April 2021, and 1 May 2020.

The second sampled wildfire happened in the area of the sub-provinces of Bayan-Uul and Bayandun on 11–28 April 2020. They are located in Northeastern Mongolia (49°15′–49°40′ N and 112°30′–113°30′ E; Figure 2(2)) in Dornod province, which is on the border between Mongolia and Russia. The burned area was calculated to be 588.35 km$^2$ at an elevation between 1100 and 1500 m. The average annual temperature of the whole range is −1–0.7 °C (the mean maximum is 20 °C in July and the mean minimum is −22 °C in January), and the average annual precipitation is 250–400 mm. Precipitation follows a bimodal distribution, with maxima in June–September and November–February. The rainiest months are July and August, with 300 mm of rain on average, and the snowiest months are December and January, with 70 mm of snow on average. Additionally, the driest periods occur from March to May and from September to November.

The first sampled area is made up of old volcano mountains and is on one side of the steppe natural zone (Figure 3). The second sampled study area is a beautiful wildland that is on the other side of the Khentii Mountains and contains a forest-steppe natural zone (experiment views shown in Figure 4). Therefore, these two sampled areas represent two different natural zones, the forest-steppe and steppe natural zones. This study estimates the results of imagined wildfires in different places and compares two different kinds of wildfires in these natural zones.

Wildfires happen in the dry seasons that occur in the spring and fall in the border zone of Mongolia. There are two kinds of legitimate wildfires in this study. One of these wildfires occurred on the Russian side. Another one occurred on the Chinese border side (Figure 2).

In the area where the fire occurred, there is a dense vegetation cover of *Stipa krylovi*, *Cleistogenes squarrosa*, *Agropyron cristatum*, *Caragana microphylla*, *Stipa krylovi*, *S. grandis*, and *Cleistogenes squarrosa*, which are typical of the Mongolian steppe. The experiment view is presented in Figure 3. In Figure 4, there is a dense vegetation cover of *Lanix sibirica*, *Pinus sylvestris*, *Betula plathyphylla*, *Carex pediformis*, *C. amgunensis*, *Iris ruthenica*, *Lathyrus humilis*, *Vicia baicalensis*, *Stipa baicalensis*, *Festuca sibirica*, *Poa attenuata*, *Filifolium sibiricum*, *Adenophora tricuspidate*, *Arenlaca sibirica*, *Amygdalus pedunculata*, *Flifolium sibiricum*, *Festuca*

*lenensis*, *Shrub-siberian tansy-fescue*, in *Henti-Ameniaca sibirica*, *Koeleria mukdenensis*, and *Lespedeza*, which are typical of the Mongolian Dagur forest steppe [70].

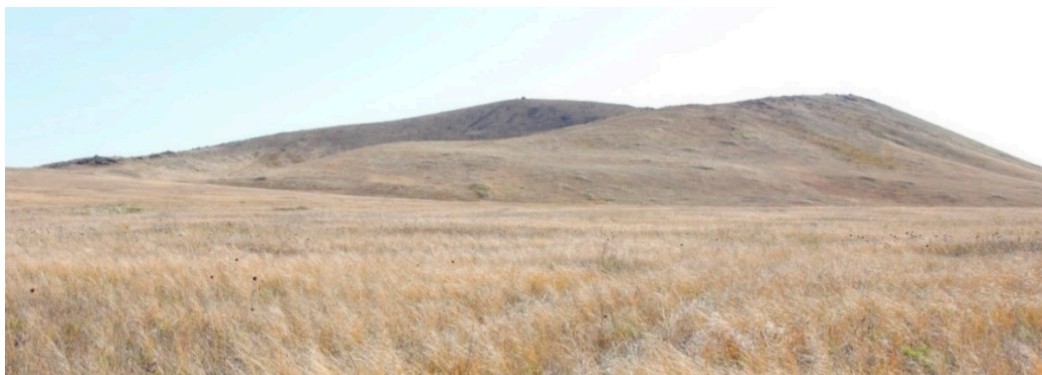

**Figure 3.** The view of the sampled area at Shiliin Bogd Mountain.

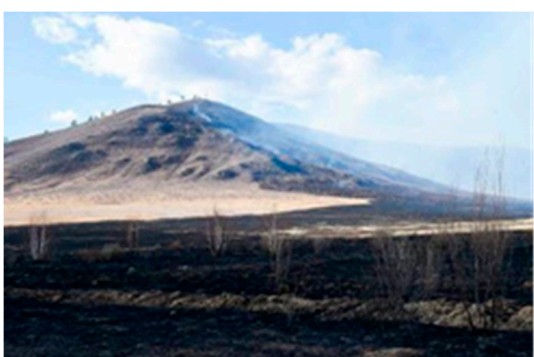 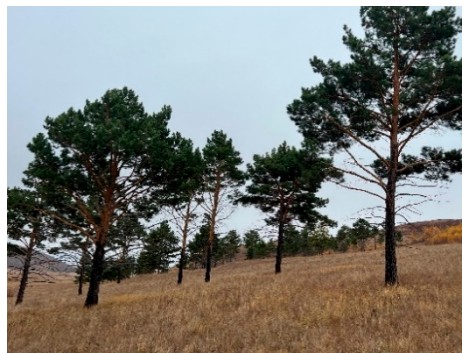

**Figure 4.** The views of the sampled areas in the sub-provinces of Bayan-Uul and Bayandun.

### 3.2. Data Collection and Processing

The Sentinel-2 satellite series has continually observed the Earth since 2015 and accumulated an enormous number of time series images. Table 1 demonstrates the characteristics of the Sentinel-2 satellite images.

**Table 1.** Spectral band characteristics and spatial resolutions of the Sentinel-2A MSI sensor. Source: ESA, 2015.

|  | Spectral Band | Spatial Resolution (m) | Center Wavelength (nm) | Band Width (nm) |
|---|---|---|---|---|
| B1 | Coastal aerosol | 60 | 443 | 20 |
| B2 | Blue | 10 | 494 | 65 |
| B3 | Green | 10 | 560 | 35 |
| B4 | Red | 10 | 665 | 30 |
| B5 | Vegetation red edge | 20 | 704 | 15 |
| B6 | Vegetation red edge | 20 | 740 | 15 |
| B7 | Vegetation red edge | 20 | 781 | 20 |
| B8 | NIR | 10 | 834 | 115 |
| B8a | Narrow NIR or NIR 2 | 20 | 864 | 20 |
| B9 | Water vapor | 60 | 944 | 20 |
| B10 | SWIR–Cirrus | 60 | 1375 | 30 |
| B11 | SWIR 1 | 20 | 1612 | 90 |
| B12 | SWIR2 | 20 | 2185 | 185 |

Sixteen cloud-free Sentinel-2 2A and 2B images were selected for this study, which were collected from April to September 2020 and 2021 (Table 2), respectively, and were acquired from ESA Sci-Hub. The Sentinel-2 satellite images were found to have better information than Landsat-8 images; therefore, Sentinel-2 was selected for image processing.

**Table 2.** Data collection of Sentinel-2 satellite images.

| Date of 1st Sampled Area | Date of 2nd Sampled Area |
| --- | --- |
| 5 April 2021 | 11 April 2020 |
| 20 April 2021 | 16 April 2020 |
| 5 May 2021 | 23 April 2020 |
| 15 May 2021 | 1 May 2020 |
| 19 July 2021 | 8 May 2020 |
| 18 August 2021 | 20 June 2020 |
| 17 September 2021 | 22 July 2020 |
| 27 September 2021 | 21 August 2020 |

In addition, a time series of satellite images was used as it can show better data from April to September of each year for the two sampled areas.

## 4. Results

The assessment and monitoring results of the wildfire burn severity and recovery process in Mongolia are analyzed in this section. Section 4.1 analyzes the defining normalized burn ratio via spectral bands. Section 4.2 identifies the burned areas in this study sampled areas; Section 4.3 analyzes the burn severity classification; and Section 4.4 monitors the recovery processes post-fire.

### 4.1. Estimation of Normalized Burn Ratio

NBR was estimated based on bands of satellite images that ranged from spring to fall. The NBR change in the images of the time series is demonstrated in Figure 5, and was calculated with Equation (1).

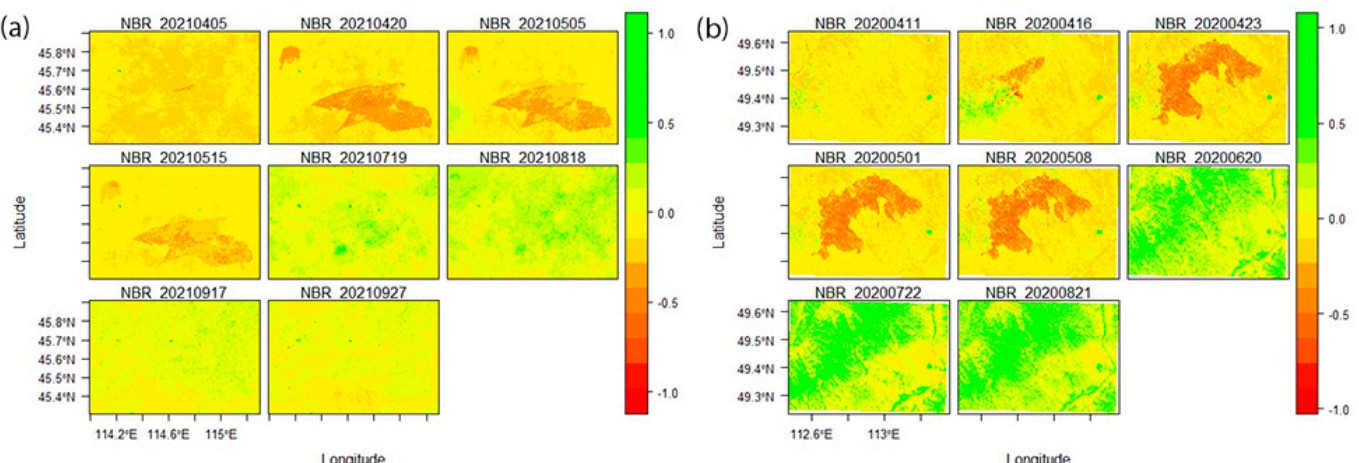

**Figure 5.** The estimated images of NBR, which are compared using time series, are (**a**) the first sampled wildfire area and (**b**) the second sampled wildfire area.

The red color indicates decreasing NBR values, whereas the green color indicates increasing values, which represent the higher and lower burned levels, respectively.

The index of the NBR is used to determine the burned area in the next section.

### 4.2. Identification of Burned Areas

NBR indices from before and after the wildfires were used to identify burned areas for the RBR index. In the first study area, satellite images were selected from 5 April 2021, and 20 April 2021. In the second study area, satellite images were selected from 11 April and 1 May 2020, respectively. There were two satellite images which were taken between 11 April and 1 May of 2020; however, because there were some clouds on the images, there are some difficulties in estimating the results for the dNBR index.

The index of the dNBR was calculated by making lower adjustments to the NBR before the fire occurred to avoid miscalculation [71]. Figure 6 demonstrates the dNBR index, which has coefficients that range between −1.37 and 1.19. It uses colored values from blue to red. Positive values range from yellow to red, which represents the estimation of the burned area. Negative values range from blue to yellow, which represents the estimation of the unburned area.

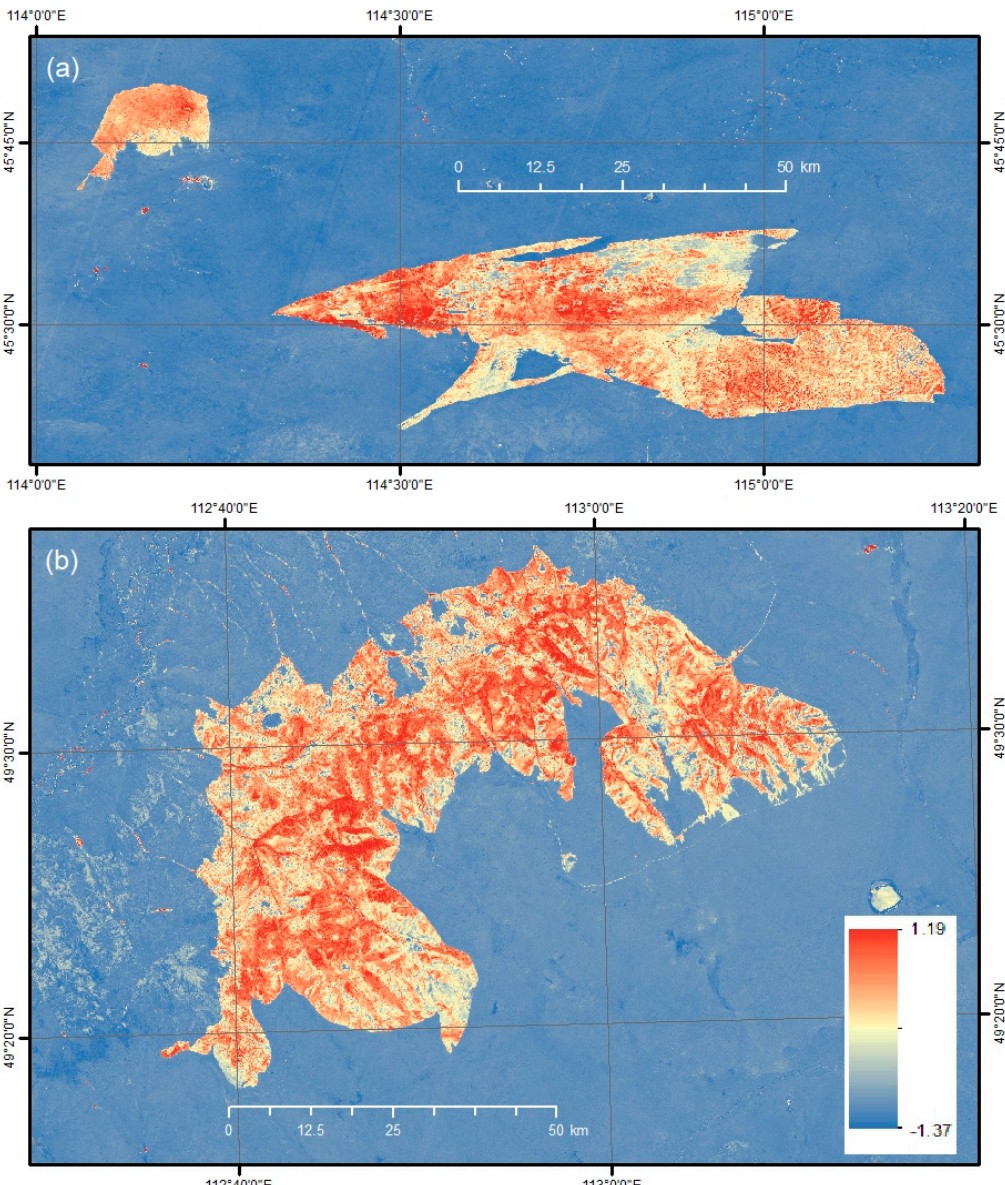

**Figure 6.** The estimation of dNBR, which is defined as damaged areas of (**a**) the first sampled wildfire area and (**b**) the second sampled wildfire area.

As shown in Figure 6, the burned area is expected to reach 1164.27 square km in the first image and 588.35 square km in the second image.

The classification of burn severity is determined in the next section.

### 4.3. Burn Severity Classification

In the third proposed result, the burn severity is classified using the pre-fire and post-fire RBR values. This helps provide aid in emergencies and assess the recovery process post-fire. The burn severity classification used here is based on that suggested by the United States Geological Survey (USGS), and it is shown in Table 3.

**Table 3.** Burn severity classification criteria table (USGS).

| Severity Level | dNBR Range (Scaled by $10^3$) | dNBR Range (Not Scaled) |
| --- | --- | --- |
| Enhanced regrowth, high (post-fire) | −500 to −251 | −0.500 to −0.251 |
| Enhanced regrowth, low (post-fire) | −250 to −101 | −0.250 to −0.101 |
| Unburned | −100 to +99 | −0.100 to +0.99 |
| Low severity | +100 to +269 | +0.100 to +0.269 |
| Moderate-low severity | +270 to +439 | +0.270 to +0.439 |
| Moderate-high severity | +440 to +659 | +0.440 to +0.659 |
| High severity | +660 to +1300 | +0.660 to +1.300 |

After the estimation of the dNBR, the RBR is calculated with Equation (3), which uses the ratio of dNBR and $NBR_{\text{pre-fire}}$. Then, the raster images of the RBR were reclassified by values from the dNBR range. This is demonstrated in Table 3. The RBR estimation raster images are shown in Figure 7. When the RBRs of this study areas are estimated in Figure 7, a comparison of the difference in the RBR between the forest-steppe and steppe areas is represented by different colors. The forest-steppe area was burned more than the steppe area.

The burned total area was divided into areas of 757.34 km$^2$ (low severity—65.00 percent), 404.57 km$^2$ (moderate-low severity—34.70 percent), and 2.36 km$^2$ (moderate-high severity—0.3 percent) in the first sampled area of the steppe natural zone, as shown in Figure 7a. The other burned total area was divided into areas of 158.75 km$^2$ (low severity—26.98 percent), 297.75 km$^2$ (moderate-low severity—50.61 percent), 131.25 km$^2$ (moderate-high severity—22.31 percent), and 0.60 km$^2$ (high severity—0.10 percent) in the second sampled area of the forest-steppe natural zone, as shown in Figure 7b. The presence of small points in the map, especially on the left side, shows the estimation of small errors in the satellite images. It was obtained at 0.07 percent in low severity (total percent was 26.98 percent) and 0.09 percent in moderate-high severity (total percent was 22.31 percent). However, these points do not cover a large area of land, which is estimated at 0.16 percent of the total area.

In the results shown in the figures, the forest and mountain areas are burned more deeply than the steppe area, which depends on wind speed.

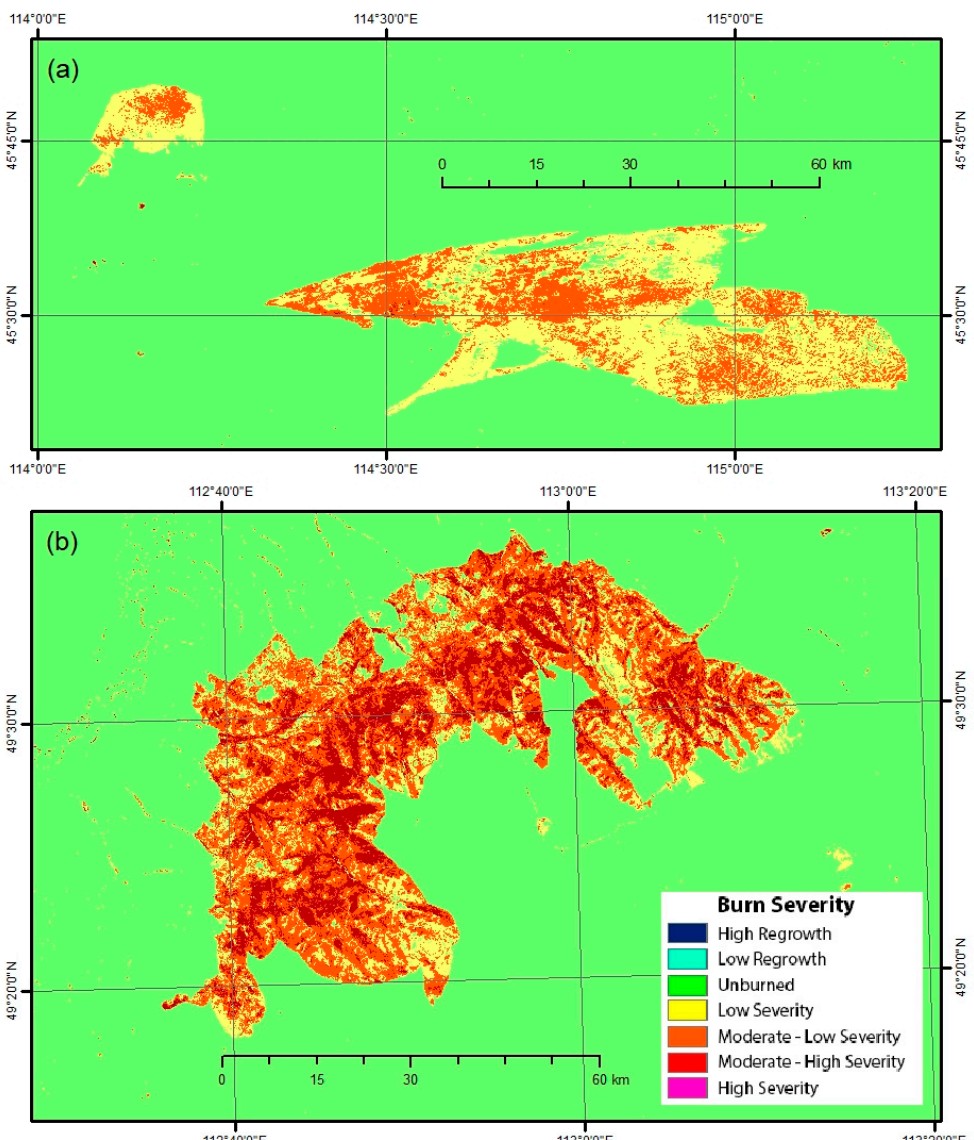

**Figure 7.** The classification of wildfire-damaged areas by burn severity is: (**a**) first sampled area; (**b**) second sampled area.

*4.4. Recovery Process after Burning*

The NDVI was used with the RED and NIR bands of the satellite images to estimate which time ranges occurred from spring to fall. The NBR change in the images of the time series is demonstrated in Figure 8 and was calculated using Equation (4).

The red color indicates decreasing NDVI values, whereas the green color indicates increasing values, which represent the higher and lower vegetation levels, respectively. Figure 8 demonstrates vegetation cover, which is compared between the two sampled study areas. The vegetation distribution level of the forest area is higher than that of the steppe area. In these two images, worse influences on nature after burning are not seen. The vegetation has recovered to be better than it was before.

In these special phenomena, vegetation grows from spring to autumn as the years pass. The vegetation dries every autumn for years, and it accumulates on the ground. Therefore, a wildfire cleans up all of the accumulated and dried vegetation by burning it. The best result is that nature heals itself, but human settlements are damaged by wildfires. However, Mongolian wildfires are not like those in Australia, Russia, and the United States, as they depend on wind speed and burn only the skin of trees. Damage due to burning does not reach 100 percent (Figure 9).

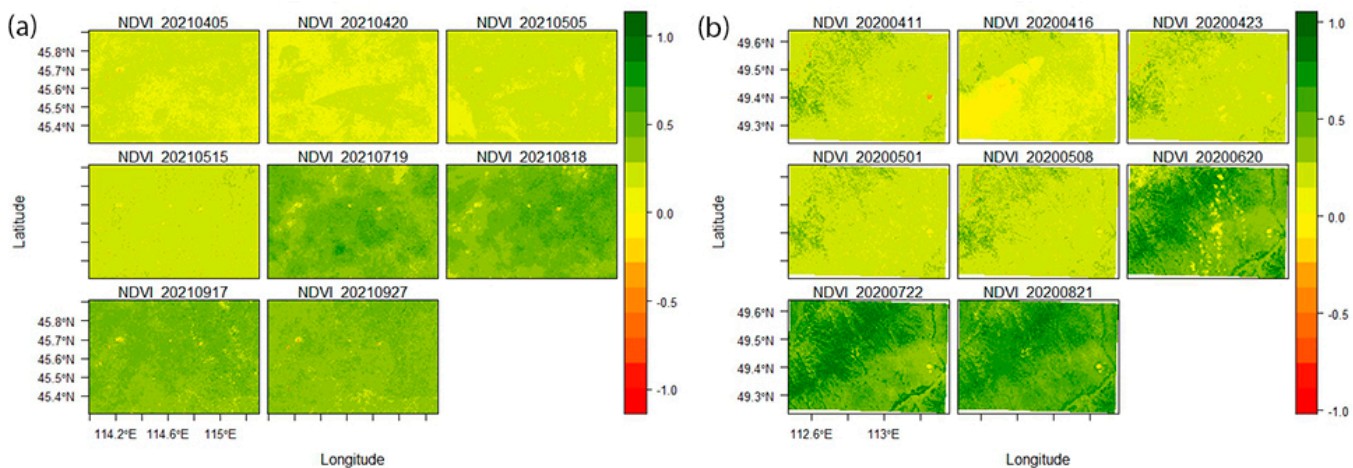

**Figure 8.** The change detection of NDVI for two sampled areas was determined with time series: (**a**) Shiliin Bogd Mountain steppe area and (**b**) forest-steppe area in the sub-provinces of Bayan-Uul and Bayandun.

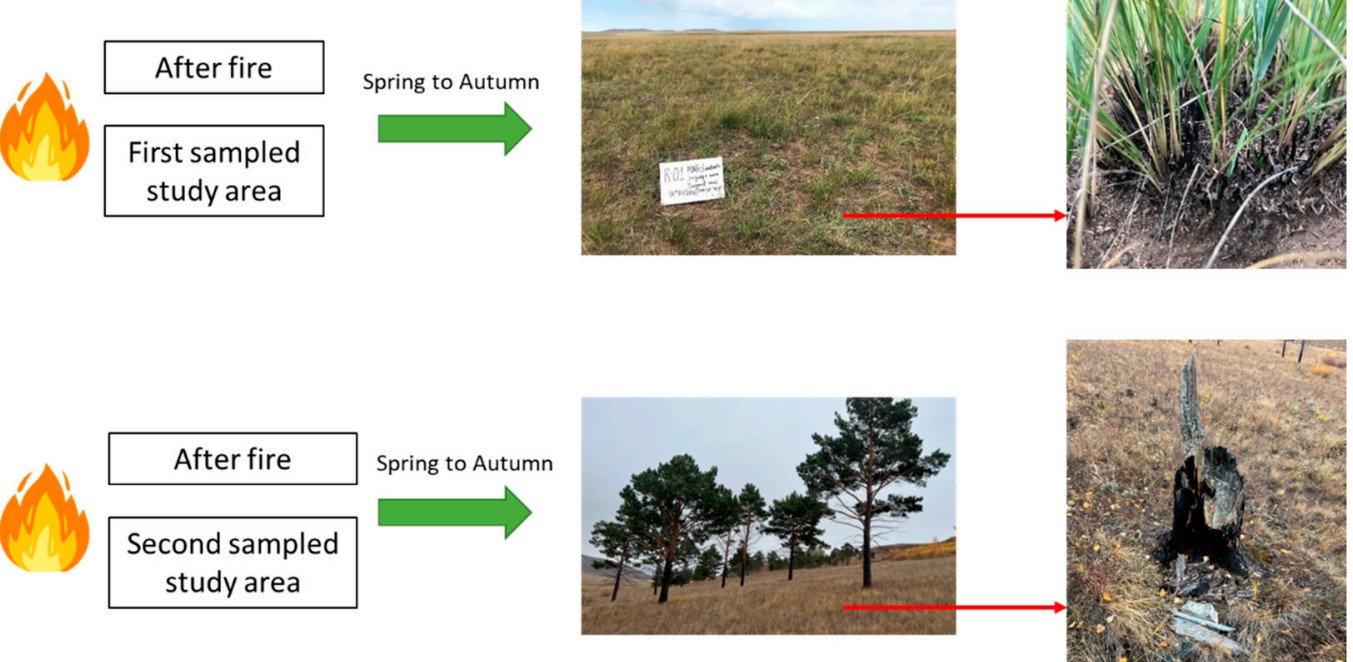

**Figure 9.** Experiment views of the recovering process at each sampled area.

In Figure 9, the top two and bottom two photographs were taken on 21 September 2021, and 12 October 2022. The comparison analysis results pre-wildfire (Figure 3) and post-wildfire (Figure 9) show that the accumulated grass is cleaned up naturally. There is no difference in vegetation growth in the forest-steppe natural zone in photographs taken 2 years later.

There is a dense vegetation cover of *Stipa krylovi*, *Cleistogenes squarrosa*, *Artemisia frigida*, *Potentilla acaulis*, and *Leymus chinensis*, which are typical of the Mongolian steppe in post-wildfire recovery. Furthermore, there is a dense vegetation cover of *Stipa baicalensis*, *Festuca lenensis*, *Potentilla strigose*, *Aster alpinus*, and *Artemisia lacinata*, which are typical of the forest steppe in post-wildfire recovery.

## 5. Discussion

First, the RBR of the wildfire severity results and the NDVI of the vegetation recovery process results were estimated. These indices were carried out during the vegetation recovery process between the spring and autumn seasons. This means that the recovering processes and collected data were a time series. RBR and NDVI data were collected at the same time. Therefore, the relationships between the raster images of the RBR and NDVI were calculated in Figures 10 and 11 for the separate natural zones, including the sampled steppe and forest-steppe areas. In this section, we will discuss these relationships, which have particular natural laws that can be seen in each figure.

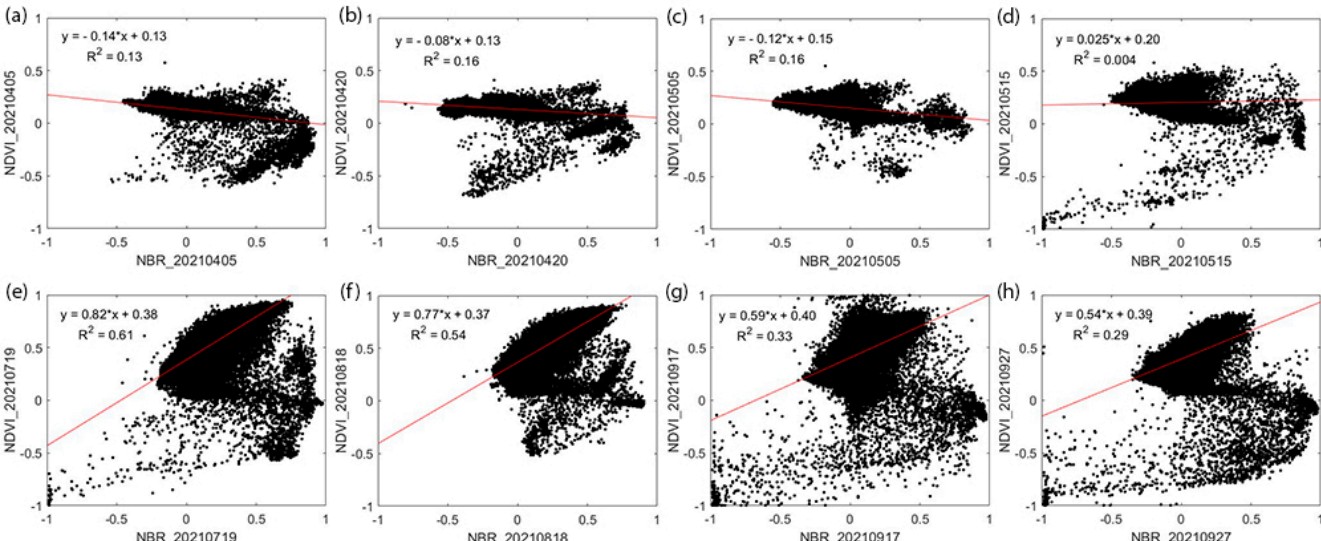

**Figure 10.** The relationships shown on scatter plots between NDVI and NBR for the recovery process in the first sampled steppe area. Each relationship estimations of the (**a**–**h**) subfigures are on each satellite image processed dates.

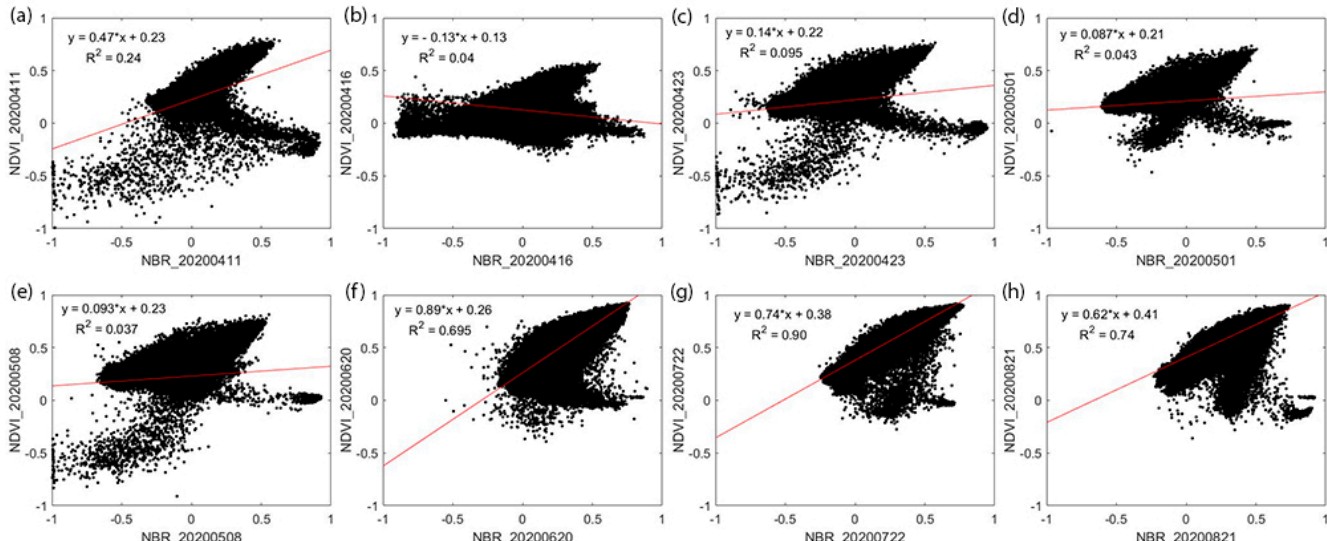

**Figure 11.** The relationships shown on scatter plots between NDVI and NBR for the recovery process in the second sampled forest-steppe area. Each relationship estimations of the (**a**–**h**) subfigures are on each satellite image processed dates.

Figure 10 illustrates the scatter plot of the recovery processes of the first sampled steppe area. The related scattering distributions are shown, which were measured on 5

April, 20 April, 5 May, 15 May, 19 July, 18 August, 17 September, and 27 September in 2021. The burn date was 18 April 2021. The coefficients of the intercepts and slopes increased, which were 0.13, 0.13, 0.15, 0.20, 0.38, 0.37, 0.40, and 0.39 and −0.14, 0.08, −0.12, 0.025, 0.82, 0.77, 0.59, and 0.54, respectively.

In addition, correlation coefficients were calculated and are shown in Figure 10, which were 0.13, 0.16, 0.16, 0.004, 0.61, 0.54, 0.33, and 0.29. The first three plots have a negative correlation with healthy vegetation cover. The exact burning date plot is shown in Figure 10b, and its correlation is lower than that of the other plots. Then, it increases until the autumn. When the last two months' vegetation becomes yellow-colored in the satellite images, the correlation decreases. The dates were in September.

Figure 11 illustrates the scatter plot of the recovery processes of the first sampled forest-steppe area. The related scattering distributions are shown, which were measured on 11 April, 16 April, 23 April, 1 May, 8 May, 20 June, 22 July, and 21 August in 2020. The burn date started on 15 April and continued to 1 May 2021. The coefficients of the intercepts and slopes increased, which were 0.23, 0.13, 0.22, 0.21, 0.23, 0.26, 0.38, and 0.41 and 0.47, −0.13, 0.14, 0.087, 0.093, 0.89, 0.74, and 0.62, respectively.

In addition, correlation coefficients were calculated and are shown in Figure 11, which were 0.24, 0.04, 0.095, 0.043, 0.037, 0.69, 0.90, and 0.74. The second plot has a negative correlation with healthy vegetation cover. The exact burning date plot is shown in Figure 11b, and its correlation is negative. Then, it increases until the autumn. When the last month's vegetation becomes yellow-colored in the satellite images, then the correlation decreases. The dates were at the end of August.

The plots in each figure demonstrate that the difference between the plots of Figure 10. is represented by lighter scattering than in the plots of Figure 11. This means that the forest-steppe natural zone experienced more severe damage due to fire than the steppe natural zone. Both recovered 100 percent by the end of the summer season.

Mongolian wildfires are increasingly caused by factors related to climate change. However, wildfires in natural zones show interesting phenomena. Site selection was demonstrated in two different natural zones, including forest-steppe and steppe areas. The wildfire severity of the forest-steppe zone was higher than the wildfire severity of the steppe zone. The wildfires in the steppe area were influenced by winds. The winds in this area are stronger than the winds in forest-stepped areas. Therefore, wildfires in the steppe burn at a low intensity.

In the recovery process, there are no effects on the sites with vegetation growing. The quality of vegetation cover grew back better than it was before the wildfire. However, the cover percentage is lighter than before the wildfire. Only tree bark and skin are affected by wildfire in forest-steppe areas. Therefore, wildfire damage is estimated to be low after the recovery process. However, if there was human property affected by the wildfire, damages would increase. Wildfires are not very harmful in Mongolia.

## 6. Conclusions

Finally, this research provides wildfire knowledge regarding the use of Sentinel-2 satellite images in the estimation of burn severity and recovery processes in Eastern Mongolia. We conclude that this research has innovated and successfully obtained new findings in the area of Mongolian wildfire research. The findings allow us to determine damage phenomena and pre-wildfire and post-wildfire differences. This research shows that wildfires do not have a large effect in Mongolia. Furthermore, this research is useful and fundamental for future wildfire studies. Future research will be based on these findings and will continue this wildfire study.

**Author Contributions:** Conceptualization, B.V. (Battsengel Vandansambuu), B.G. and Y.B.; methodology, B.G. and N.C; validation, B.V. (Battsengel Vandansambuu), N.B. and O.B.; formal analysis, B.G. and S.B.; investigation, B.V. (Battsengel Vandansambuu), and B.V. (Batbayar Vandansambuu); resources and curation, N.C., B.V. (Batbayar Vandansambuu), and M.-E.J.; writing—original draft preparation, B.G. and F.W.; writing—review and editing, F.W.; visualization, N.C. and S.B.; supervi-

sion, B.V. (Battsengel Vandansambuu); project administration, B.V. (Battsengel Vandansambuu), and B.G. All authors have read and agreed to the published version of the manuscript.

**Funding:** This research was funded by the National Natural Science Foundation of China (41861021) and the Mongolian Foundation for Science and Technology. Also, we have another project named "Wildfire Monitoring of Natural Disasters and Its Risk Assessment Using Remote Sensing Methods in Mongolia" (APSCO/PO&DS/1st BATCH OF DSSP APPLICATION/IMP_C_008), which was funded by the Asia-Pacific Space Cooperation Organization (APSCO). The National University of Mongolia (NUM) supports the implementation of the projects (P2022-4261 and P2022-4398) in the field.

**Institutional Review Board Statement:** Not applicable.

**Informed Consent Statement:** Not applicable.

**Data Availability Statement:** Not Applicable.

**Acknowledgments:** We would like to thank the Copernicus Open Access Hub of ESA, the EarthExplorer of USGS, and the Data Sharing Service Platform (DSSP) of APSCO for providing accessible data products. We would also like to thank the National University of Mongolia (NUM) for all its support. We are highly grateful to the anonymous editors and reviewers for their valuable suggestions and comments that significantly improved the manuscript's quality.

**Conflicts of Interest:** The authors declare no conflict of interest.

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
