# Peer review of "Assessment of Burn Severity and Monitoring of the Wildfire Recovery Process in Mongolia"

_fire, doi:10.3390/fire6100373_

Round 1

Reviewer 1 Report

The work in rather interesting and promising but it is hard to read because of many errors. Please consult a translator rather than using an automatic translator

The conclusion does not follow from the work

English is hard to read and lot of sentences sounds colloquial. 

Reviewer 2 Report

The Assessment of Burn Severity and Monitoring of Recovery 3 Process of Wildfire in Mongolia

Dear Authors

The basic science of this paper is conducted not in a good way and is not an appropriate standard.  The author and his team write this paper according to the journal's scope and modern trends. I am glad to review this paper because it’s related to my research. I have seen thousands of papers related to this topic and the study area has been published in well-reputed journals. If the authors want to publish this study, they should provide some novelty or enhance the significance of the research. I am going to recommend major changes in this paper. I hope the authors will follow our comments enhance their study and resubmit to this journal. 

Title

The title is fine and according to the study

Abstract

The abstract is appropriate according to journal criteria. The abstract does not reflect the significance/ purpose of this research. The authors should revise the whole abstract. The abstract section is also very lengthy.

Introduction

Check the language of the first two paragraphs of the introduction section. This is not the novel way, I have seen many papers related to this topic. So, the authors should replace this word with a suitable word.

What new contributions and understanding does your research provide?

How does your research complement and support the key hypotheses from the published literature on this subject and the modeling applications?

The authors should rewrite the Introduction section, extract research questions and useful information from the literature, and point out the shortcomings of previous studies, thus leading to the article's environmental significance and purpose.

Authors should particularly pay attention to the literature review which should be more critical. The authors should detail the methodological novelties with the vast amount of existing literature in this area. The author should read and cite these latest articles in the revised version

The author should clearly explain the main objectives of this study with a central hypothesis which is missing in the Introduction.

The introduction section is very lengthy

The author should follow the IMRAD rule

The author should restructure this paper.

I suggested some papers. You can follow and rewrite this paper.

2. Analyzing methods for wildfire monitoring

Remove the figure 1.

Figure 2 caption is not appropriate. The author should modify the caption and explain all figures in the caption

Results

Figure 5: The author should paste horizontally

Check all figures

Discussions

It's ok

Conclusion

The conclusion section is lengthy. The author removes some text from the manuscript. And add how this study is suitable for SDGS

I hope, the authors will improve this study and resubmit it in this journal.

Figures are not appropriate according to journal standard

Best Regards

Reviewer 3 Report

The manuscript is a very well-designed work and I can say that it is academically valuable. The effects of fires and their aftermath may vary in each region. The authors have successfully demonstrated this process in their study area. However, the current version of the manuscript contains several deficiencies. I believe it will be useful to list them after they are fixed.

The shortcomings I have identified in the manuscript that need to be addressed are as follows:

1. The first sentence in the introduction, between lines 39-41, requires references to recent fire studies. I found some relevant papers published in 2023:

Cosgun, U.; Coşkun, M.; Toprak, F.; Yıldız, D.; Coşkun, S.; Taşoğlu, E.; Öztürk, A. Visibility Evaluation and Suitability Analysis of Fire Lookout Towers in the Mediterranean Region, Southwest Anatolia/Turkey. Fire 2023, 6, 305. https://doi.org/10.3390/fire6080305

Ntinopoulos, N.; Sakellariou, S.; Christopoulou, O.; Sfougaris, A. Fusion of Remotely-Sensed Fire-Related Indices for Wildfire Prediction through the Contribution of Artificial Intelligence. Sustainability 2023, 15, 11527. https://doi.org/10.3390/su151511527

Tsioras, P.A.; Giamouki, C.; Tsaktsira, M.; Scaltsoyiannes, A. What the Fire Has Left Behind: Views and Perspectives of Resin Tappers in Central Greece. Sustainability 2023, 15, 9777. https://doi.org/10.3390/su15129777

Yilmaz, O.S., Acar, U., Sanli, F.B. et al. Mapping burn severity and monitoring CO content in Turkey's 2021 Wildfires, using Sentinel-2 and Sentinel-5P satellite data on the GEE platform. Earth Sci Inform 16, 221–240 (2023). https://doi.org/10.1007/s12145-023-00933-9

2. The paragraph between lines 50-61 needs additional references regarding the various factors discussed in the cited papers. The authors should find them in the literature.

3. The two sentences between lines 62-64 should be combined since the first sentence does not mention satellite data and seems unnecessary in its current form.

4. The sentence in line 72 is unclear. Please clarify its meaning. Additionally, the paragraph between lines 72-80 needs to be rewritten for better clarity, as it currently has multiple interpretations for readers.

5. Include important forest fires affecting Turkey and Greece in addition to the fires mentioned in lines 88-93.

6. Line 122: Avoid using acronyms without clear meanings. Ensure all acronyms used in the paper are explained or spelled out.

7. Line 132: Ensure that Figure 1 is referenced in the main text before it appears for the first time.

8. Line 183: Clarify what "200.6 mm" represents. Is it related to precipitation?

9. Lines 200-206: Provide detailed information about the plant species covering the study area at the beginning of the study. Different plant species may have distinct post-fire regeneration processes, making this information crucial.

10. Line 255: If you use graticules (congratulations on choosing to use this.), remove North arrows from Figure 6 and others, as it is a cartographic rule.

11. Line 273: Ensure that the colors representing high and low regrowth in the map are distinguishable.

12. Line 273: Address the presence of small points in the map, especially on the left side, which may resemble burned areas. Explain whether these are indeed burned areas and why they might not have expanded or if they are something else.

13. Lines 281-287: Specify if the small areas mentioned above are included in the values provided here and quantify their total coverage.

14. Line 293: Provide the exact times when the images were taken, rather than relying on seasonal conversion.

15. Line 313: Address the presence of different plant species in the area that should have been analyzed initially.

I hope these revisions and clarifications help improve the manuscript's quality and readability. 

No comments.

Round 2

Reviewer 1 Report

You did a good job, thanks for listening to the recommendations. One moment left.

At lines 34 and 301 where total burned area of the forest-steppe natural zone is describes numbers are little strange: 26.90+50.61+21.31+0.1 = 98.92%

Where is remaining 0.8% and there are two moderate-high areas at line 34 instead of moferate-high and high.

It seems that it is for "small points in the map, especially on the left side, which shows the estimation of small errors on the satellite images". 

Either do not include them in the total area, or specify like "the remaining 0.8% of the burnt area is a satellite data error".

Reviewer 2 Report

agree with the revised version

Author Response

There is no any comments. Thanks